# Factors That Influence Consumers' Sustainable Apparel Purchase Intention: The Moderating Effect of Generational Cohorts

**Pei-Hsin Lin *** and **Wun-Hwa Chen**

Department of Business Administration, National Taiwan University, Taipei 106, Taiwan; andychen@ntu.edu.tw
* Correspondence: jilllin329@gmail.com

**Abstract:** The circular economy is one of the crucial issues in fashion because the fashion industry is a major global polluter. Many consumers are adopting a more sustainable lifestyle and it shows in their buying preferences and behaviors. This study aims to predict sustainable fashion apparel consumption using an extended version of the belief–attitude–intention framework, by investigated the moderating effect of generational cohorts. Particularly, the study emphasizes the rental apparel, second-hand apparel, and recycled apparel markets. Survey data were collected from 135 Generation X consumers, 134 Generation Y consumers, and 139 Generation Z consumers in Taiwan. Structural equation modeling and the bootstrapping method were applied to test the hypothesized relationships. The findings determined environmental consciousness, perceived value, and perceived risk as key predictors of consumers' sustainable apparel purchase intentions. The findings also showed that the generational cohort negatively moderated the relationship between environmental consciousness and sustainable apparel purchase intentions. Therefore, fully understanding consumers' purchase intentions regarding sustainable apparel is an indispensable topic for both academia and industry in a circular environment. Moreover, the fashion industry should concentrate more on promoting sustainability and ecologically friendly apparel products as well as developing multi-generational marketing strategies.

**Keywords:** fashion industry; sustainable apparel; environmental consciousness; perceived value; perceived risk; purchase intention; generational cohort

## 1. Introduction

The fashion industry is widely believed to be the second largest polluting industry on the planet, and the environmental damage is increasing as the industry grows [1,2]. According to Niinimäki et al. [3], garment production is responsible for around 20% of industrial wastewater pollution from dyeing and finishing textiles, 8–10% of carbon dioxide emissions, and more than 92 million tonnes of waste produced per year. Meanwhile, fast fashion has caused a considerable rise in the quantity of garments produced and thrown away [4]. Recently, these problems throughout the textile and fashion supply chains have appealed to consumers' growing concerns and demands for sustainable products made according to ecological and social principles, and thus a growing number of firms are committing to minimizing their detrimental impacts on ecosystems and societies [5–7]. However, consumers' understanding of sustainable apparel are often vague, and their buying decisions between sustainable and non-sustainable apparel often depend on aesthetic, functional, and financial benefits, resulting in low involvement of consumers' sustainable apparel consumption [8–10]. For that reason, it is critical to clearly identify the important factors affecting sustainable apparel purchase intentions.

In the field of sustainable fashion consumption, renting and buying second-hand or recycled clothing are considered as sustainable options, and there is an increasing body of

literature that recognizes the relationship between consumers' environmental consciousness, perceived values, perceived risks, and purchase intentions [11–13]. In addition, recent evidence confirms that consumers of different generations have different sustainable consumer behaviors due to their different living backgrounds and environments [14–17]. So far, however, much uncertainty still exists about the relationship between environmental consciousness and purchase intention in the sustainable apparel market, and it is still not known how different generations moderate sustainable apparel purchase intentions. Therefore, this study attempts to show how an individual's environmental consciousness affects their sustainable apparel purchase intention through perceived values and perceived risks, and to recognize the moderating roles of the generations. In this study, sustainable fashion apparel covers rental clothing, second-hand clothing, and recycled clothing, and the targeted sample group consists of Generation X, Generation Y, and Generation Z. The study presumes that consumers who have high environmental consciousness may present a stronger behavioral intention to purchase sustainable fashion apparel, and a moderating effect of generational cohort may exist in the consumers' sustainable apparel purchase intention.

This study makes an important contribution to the circular fashion industry and provides valuable insights for fashion brands or retailers to understand the different generations' sustainable apparel consumption preferences and behaviors regarding different sustainable apparel products. The rest of the study is organized as follows: Section 2 presents the literature review on circular fashion, environmental consciousness, perceived value, perceived risk, and generational cohort theory; Section 3 introduces the research methodology and sample profile; Section 4 displays the hypothesis testing results for the three types of sustainable apparel, and in particular, the moderating effects of generational cohorts are presented; Section 5 examines the empirical findings in detail; Section 6 summarizes the main findings and limitations of the current study, with future research recommendations.

## 2. Theoretical Framework

### 2.1. Circular Fashion

The circular economy aims to change the linear material flow, which directs many manufacturing operations, into closed-loop models that emphasize that resource and residual waste are shared, repaired, reused, or recycled [18,19]. A circular model for fashion generates green products and services for customers, while contributing to a prosperous fashion industry and environmental regeneration [20,21]. Given the consumer's growing awareness of sustainable consumption, there has been an increasing amount of literature on alternate models of clothing consumption [22–25]. Kim et al. [12] classified sustainable fashion products into three types: second-hand clothing, upcycled clothing, and recycled clothing. Machado et al. [26] demonstrated that the second-hand clothing business model is a way of decreasing resource use where waste in the form of used goods is reused, and product lifespans are extended through transferring ownership. Additionally, Clube and Tennant [27] and Shrivastava et al. [28] showed that online rental of used clothing is an emerging business model that bolsters circular fashion practices linked to environmental and economic sustainability. As a result, the circular fashion industry aims to promote the re-use and recycling of clothing. In the current study, we investigated the three different types of sustainable apparel consumption: rental clothing, second-hand clothing, and recycled clothing.

### 2.2. Environmental Consciousness

Environmental consciousness is considered as a complex concept built upon cognitive, attitudinal, and behavioral components as well as environmental knowledge [29,30]. Accordingly, environmental consciousness is an element of the belief system that contributes to specific mental influences linked to one's tendency to participate in the eco-friendly behavior regime [31]. To date, many recent studies have highlighted that environmental

consciousness has become a vital component of the consumer decision-making process in the sustainable consumption context [32–35]. For example, Kautish and Sharma [36] and Zhang et al. [37] validated the relationships among environmental consciousness, perceived values, and behavioral intentions for green products. Wang et al. [38] revealed that perceived value regarding quality and price was a mediator of environmental consciousness and green purchase behaviors. Similarly, Hasan et al. [39] confirmed the positive relationship between environmental concerns, consumer attitudes, and a willingness to purchase organic cotton clothing. Souza et al. [40] posited that environmental consciousness, perceived values, and risk factors are key predictors of environmental purchase behavior. Szabo and Webster [41] indicated that environmental beliefs have positive effects on perceived value and negative effects on perceived risk. Therefore, these subsequent hypotheses are proposed:

**H1**: *Environmental consciousness is positively associated with perceived value.*

**H2**: *Environmental consciousness is negatively associated with perceived risk.*

*2.3. Perceived Value and Perceived Risk*

Perceived value is a customer's personal assessment of a product's utility [42]. According to the consumption values theory, consumers' product choice behaviors are established on five perceived values: functional, emotional, social, epistemic, and conditional [43]. In the textile and apparel domain, a multidimensional perspective of perceived value has been widely adopted to investigate consumers' sustainable clothing purchase behaviors. Bielawska and Grebosz-Krawczyk [44] analyzed data from 496 Polish consumers and concluded that emotional, conditional, and environmental values had significantly positive influences on purchase behavior regarding eco-friendly clothing. Baek and Oh [45] studied consumers' adoption intentions regarding fashion apparel rental services and revealed that functional, economic, and emotional values boost attitudes and intentions. Chun et al. [46] suggested that social value, emotional value, functional value, economic value, and eco value all have positive influences on behavioral intentions regarding recycled fashion products. Chi et al. [47] identified the perceived green value of recycled polyester-made athleisure apparel products as a multidimensional construct, including functional value, social value, emotional value, conditional value, and epistemic value. Considering this evidence, it seems that perceived value positively affects behavioral intention regarding sustainable apparel products or services. Therefore, the following hypothesis is proposed:

**H3**: *Perceived value is positively associated with sustainable apparel purchase intention.*

In previous studies on consumer decision-making behaviors, researchers conceptualized perceived risk as customers' beliefs about the uncertainty and potential negative outcomes of purchasing a product or service [48–50]. To date, several studies have highlighted various types of risk that are associated with consumers' purchase intentions, such as time, financial, social, psychological, physical, and performance risk [51–55]. In the case of online fashion rental services, Lee et al. [13] demonstrated that financial risk, performance risk, and social risk have a negative effect on usage intention. Similarly, Yoo et al. [56] proposed that key obstacles to consumer purchase intentions regarding upcycled apparel were social, financial, and performance risks. Moreover, hygiene concerns are suggested to be a strong barrier to the purchase intention of sustainable clothes [24,57–59]. Park and Choo [60] and Kim et al. [12] proved that sanitary risk negatively affects the intention to purchase second-hand, upcycled, and recycled clothing. Collectively, these studies suggest that perceived risk negatively influences behavioral intentions regarding sustainable apparel products or services. Therefore, the following hypothesis is proposed:

**H4**: *Perceived risk is negatively associated with sustainable apparel purchase intention.*

### 2.4. Generational Cohort Theory

Mannheim [61] proposed the generational cohort theory (GCT) and stated that a generational cohort is a set of individuals born within a particular span of time who share a similar age and stage of life. More specifically, a generation refers to a cohort of people who have comparable age and experience similar social, economic, political, and cultural events [62–64]. As a result, each generation would inherit a collective consciousness and develop unique values, belief systems, peer personalities, and behavioral tendencies [65]. According to Dimock [66], the generational cohort can be divided into five age groups: the Silent Generation (born 1928–1945), Baby Boomers (born 1946–1964), Generation X (born 1965–1980), Generation Y (also known as Millennials; born 1981–1996), and Generation Z (born 1997–2012). With members of Generation X and Y representing the largest percentage of today's workforce, and the oldest members of Generation Z are entering the workforce, the targeted sample of this study was Generations X, Y, and Z. In essence, Generation Xers are those born before the widespread adoption of digital technology; Generation Yers are also known as digital natives and technology is part of their everyday lives; Generation Zers have been exposed to the internet, to social networks, and to mobile systems.

Thus far, many previous studies in the field of marketing have demonstrated that an individual's decision-making processes [67,68], online shopping orientations [69–71], social media and technology usage [72,73], brand engagement and loyalty [74], and sustainable consumption behaviors [17,75] could be modified by that individual's generational identity. Much of the current literature on sustainable fashion apparel consumption pays particular attention to generational differences. Kim [76] pointed out that there are significant differences in the cognitive, affective, and behavioral responses to fashion luxury products between Millennials and Baby Boomers. Gazzola et al. [15] confirmed that the younger generations today are more aware of sustainability and the circular economy. Liang and Xu [77] found that Generation X demonstrated a high resistance to second-hand clothing products, while younger generations held stronger perceived values and purchase intentions regarding second-hand clothing than their older counterparts. Overall, the evidence reviewed here seems to suggest a moderating role for generational cohorts. Therefore, the subsequent hypotheses are proposed:

**H5**: *Generational cohorts moderate the direct effects of environmental consciousness on (a) perceived value, and (b) perceived risk.*

**H6**: *Generational cohorts moderate the direct effects of (a) perceived value, and (b) perceived risk on sustainable apparel purchase intention.*

### 2.5. Conceptual Model

Drawing on the belief–attitude–intention framework and the generational cohort theory, the purpose of this study is to explore the impact of environmental consciousness on perceived value, perceived risk, and sustainable apparel purchase intentions, and to investigate how generational differences moderate the association between environmental consciousness and sustainable apparel purchase intention. The conceptual model is displayed in Figure 1, which encompasses the above-mentioned five dimensions.

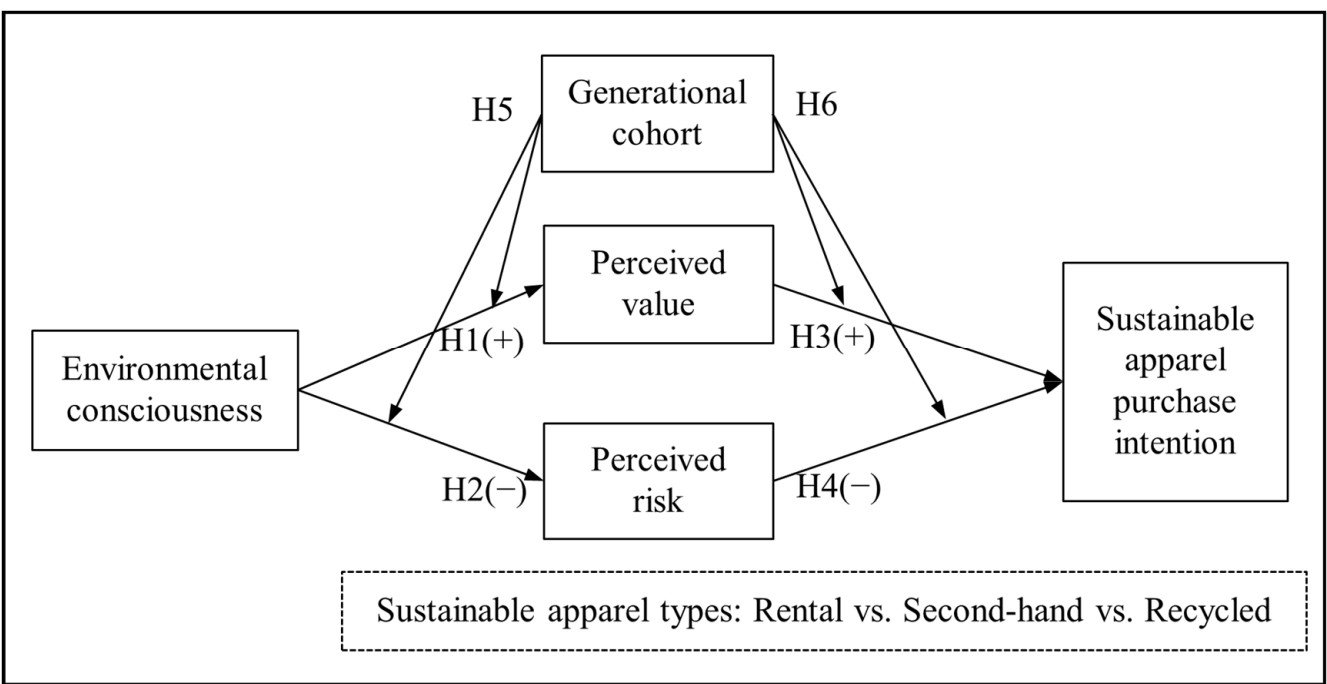

**Figure 1.** Conceptual model of this study.

## 3. Methods

### 3.1. Sample and Procedure

The current research investigated the determinants of consumers' sustainable apparel purchase intentions, and further examined the moderating role of generational cohorts. A survey approach was used to conduct this quantitative study, with the samples for the study drawn from Taiwan. Individuals aged below 56 were the targeted sample in this study, and the sample size of 200 or more for this study was determined because a minimum sample size of 200 is recommended for survey studies of behavior/cognition [78]. This study employed the SurveyCake online survey platform to collect the target samples. SurveyCake (https://www.surveycake.com/ (accessed on 3 September 2021)) is a market research platform that surveys internet users, and was used because it contains millions of users and offers representative sampling techniques where questionnaires can be sent to specific demographic groups to obtain accurate audience samples and target numbers. Participants who completed all scales for three types of sustainable clothing received TWD 100 as an incentive. A total of 408 valid responses were obtained between October and December 2021 and used for statistical analysis.

### 3.2. Measurement and Analytic Method

The survey started with a brief outline of the investigation's purpose and illustrations of three types of sustainable apparel: rental clothing, second-hand clothing, and recycled clothing. The questionnaire included two sections: (1) demographic: gender, age, marital status, monthly discretionary income, average monthly apparel expenditure and sustainable apparel purchase experience; (2) survey: items measuring environmental consciousness, perceived value, perceived risk, and sustainable apparel purchase intention. All multi-item instruments were measured utilizing 5-point Likert scales (1 = Strongly disagree, 3 = Neutral, 5 = Strongly agree) and were well-established measurements adapted from previous studies. Environmental consciousness was evaluated using three items from Wang et al. [38]. To access the perceived value in purchasing sustainable apparel products, five items related to functional, emotional, social, epistemic, and environmental value were measured referring to Kim et al.'s [12] and Bielawska and Grebosz-Krawczyk's [44] studies. As for perceived risk, four items measuring social, financial, performance, and sanitary risk

were adopted from Yoo et al. [56] and Kim et al. [12]. Items measuring sustainable apparel purchase intention (3 items) were taken from Lee et al. [13] and Yoo et al. [56].

For data analysis, descriptive analysis, confirmatory factor analysis (CFA), structural equation modeling (SEM) analysis, and moderation analysis were used. SPSS software was utilized in the study to analyze demographic data and moderation effects. A confirmatory factor analysis (CFA) was used to evaluate how well the survey results matched the hypothesized measurement model. CFA enables researchers to systematically test specific prior hypotheses about the structure underlying survey results and to compare alternative measurement models with respect to explanatory power; this makes CFA a valuable tool for theory testing and building [79]. In addition, the hypotheses were tested by applying SEM, which is the most commonly used estimation method for robust analysis of data in the behavioral and social sciences [80]. CFA and SEM were performed adopting AMOS software.

### 3.3. Sample Profile

The sample profile shown in Table 1 was obtained via a descriptive analysis of the data. The sample consisted of 33.1% (*n* = 135) generation X, 32.8% (*n* = 134) generation Y, and 34.1% (*n* = 139) generation Z, indicating that the sample sizes for the three cohorts were equivalent. Of the 408 respondents, 245 (60%) were female and 163 (40%) were male. Most respondents in Generation X were married (67.4%) and the Generation Z respondents were predominantly (77.7%) single. As for their discretionary income, more than 60% of Generation X respondents had over TWD 20,000 a month, while the Generation Z respondents indicated a monthly discretionary income of less than TWD 20,000 (60.4%). In terms of average monthly apparel expenditure, between TWD 1001 and TWD 4000 was the most common, followed by less than TWD 1000. For the sustainable apparel purchase experience, most respondents had purchased second-hand and recycled clothing but had never used clothing rental services.

**Table 1.** Sample profile.

| Variable | | Generation X (Ages 41–56) | Generation Y (Ages 25–40) | Generation Z (Ages 9–24) |
|---|---|---|---|---|
| | | Number (%) | Number (%) | Number (%) |
| Gender | Male | 58 (43.0%) | 54 (40.3%) | 51 (36.7%) |
| | Female | 77 (57.0%) | 80 (59.7%) | 88 (63.3%) |
| Marital status | Married | 91 (67.4%) | 57 (42.5%) | 31 (22.3%) |
| | Single | 44 (32.6%) | 77 (57.5%) | 108 (77.7%) |
| Monthly Discretionary Income | <TWD 10,000 | 13 (9.6%) | 25 (18.7%) | 42 (30.2%) |
| | 10,001–20,000 | 37 (27.4%) | 35 (26.1%) | 42 (30.2%) |
| | 20,001–30,000 | 35 (25.9%) | 35 (26.1%) | 38 (27.3%) |
| | 30,000–40,000 | 29 (21.5%) | 21 (15.7%) | 14 (10.1%) |
| | >TWD 40,000 | 21 (15.6%) | 18 (13.4%) | 3 (2.2%) |
| Average monthly apparel expenditure | <TWD 1000 | 39 (28.9%) | 33 (24.6%) | 43 (30.9%) |
| | 1001–4000 | 63 (46.7%) | 71 (53.0%) | 65 (46.8%) |
| | 4001–7000 | 21 (15.6%) | 16 (11.9%) | 18 (12.9%) |
| | 7001–10,000 | 9 (6.7%) | 6 (4.5%) | 12 (8.6%) |
| | >TWD 10,000 | 3 (2.2%) | 8 (6.0%) | 1 (0.7%) |
| Ever use clothing rental services? | Yes | 60 (44.4%) | 60 (44.8%) | 57 (41.0%) |
| | No | 75 (55.6%) | 74 (55.2%) | 82 (59.0%) |
| Ever buy second-hand clothing or resale used clothing? | Yes | 84 (62.2%) | 92 (68.7%) | 83 (59.7%) |
| | No | 51 (37.8%) | 42 (31.3%) | 56 (40.3%) |
| Ever buy recycled clothing? | Yes | 89 (65.9%) | 90 (67.2%) | 76 (54.7%) |
| | No | 46 (34.1%) | 44 (32.8%) | 63 (45.3%) |

## 4. Results

### 4.1. Reliability and Validity

A confirmatory factor analysis was utilized to examine the reliability and validity of the measurement model. Table 2 exhibits mean and estimated factor loadings (FL) for each item and Cronbach's α, composite reliabilities (CR), and average variance extracted (AVE) for each dimension. All FL values ranged from 0.709 to 0.914, and the AVE for all constructs was between 0.512 and 0.815, surpassing the 0.5 cutoff value. The value for CR should exceed 0.7. The coefficient values of CR in this study were between 0.758 and 0.930. All estimated FL, AVE, and CR values met the relevant criteria, providing significant evidence of convergent validity [81]. In addition, Cronbach's α was used to evaluate the internal reliability of the measurements. The Cronbach's α values for all dimensions were in the range of 0.825–0.928, surpassing the suggested threshold of 0.7 [82]. Thus, these results reveal that the measurement of this study was reliable and valid. From the table, it can be seen that the highest AVE, CR and Cronbach's α values were for the sustainable apparel purchase intention dimension (AVE = 0.512, CR = 0.758, Cronbach's α = 0.825), while the lowest AVE, CR and Cronbach's α values were for the environmental consciousness dimension (AVE = 0.815, CR = 0.930, Cronbach's α = 0.928).

**Table 2.** Results of reliability and validity analyses.

| Dimension | Items | Mean | FL | Statistics |
|---|---|---|---|---|
| Environmental consciousness | 1. The balance of nature is very delicate and can be easily upset. | 4.219 | 0.709 | CR = 0.758, AVE = 0.512, Cronbach's α = 0.825 |
| | 2. I have switched products for ecological reasons. | 4.127 | 0.711 | |
| | 3. When I have a choice between two equal products, I purchase the one less harmful to other people and the environment. | 4.153 | 0.725 | |
| Perceived value | 1. This sustainable clothing is well made and worth the money. | 3.784 | 0.874 | CR = 0.917, AVE = 0.690, Cronbach's α = 0.889 |
| | 2. Purchasing this sustainable clothing makes me feel good. | 3.699 | 0.909 | |
| | 3. Purchasing this sustainable clothing can give its owner social approval. | 3.599 | 0.801 | |
| | 4. This sustainable clothing offers uniqueness. | 3.756 | 0.815 | |
| | 5. This sustainable clothing helps save resources. | 3.953 | 0.744 | |
| Perceived risk | 1. This sustainable clothing would not be durable. | 3.221 | 0.840 | CR = 0.890, AVE = 0.668, Cronbach's α = 0.889 |
| | 2. This sustainable clothing is unlikely to be hygienic. | 3.600 | 0.788 | |
| | 3. I would not feel comfortable wearing this sustainable clothing in public. | 3.421 | 0.783 | |
| | 4. I think it is not worthwhile to spend money on this sustainable clothing. | 3.210 | 0.857 | |
| Sustainable apparel purchase intention | 1. I am willing to visit a store that sells this sustainable clothing. | 3.662 | 0.883 | CR = 0.930, AVE = 0.815, Cronbach's α = 0.928 |
| | 2. I am willing to visit the website of this sustainable clothing. | 3.544 | 0.914 | |
| | 3. I am willing to recommend this sustainable clothing to others. | 3.653 | 0.911 | |

Sustainable clothing corresponds to rental clothing, second-hand clothing, and recycled clothing. The mean is the average of the numbers of the three types of apparel. Significance level: $p < 0.05$; $p < 0.01$; $p < 0.001$.

### 4.2. Hypothesis Verification

Structural equation modeling (SEM) was carried out to examine the interrelationships between environmental consciousness, perceived value, perceived risk, and sustainable apparel purchase intention using the AMOS software. Table 3 shows the main research hypotheses for overall sustainable apparel; except for H2, all other research hypotheses were supported. In detail, environmental consciousness had a significant positive impact on perceived value ($\beta = 0.364$, $p < 0.001$), and perceived value had a significant positive impact on sustainable apparel purchase intention ($\beta = 0.919$, $p < 0.001$), whereas perceived risk had a significant negative impact on sustainable apparel purchase intention ($\beta = -0.122$, $p < 0.001$). No significant influence was found between environmental consciousness and perceived risk ($\beta = 0.014$, $p > 0.05$). In terms of the differences between apparel types, for rental and second-hand apparel, H1, H3, and H4 were supported, but H2 was not supported. As for recycled apparel, H1 and H3 were supported but H2 and H4 were not supported, indicating that perceived risk did not significantly mediate the relationship between environmental consciousness and recycled apparel purchase intention. Moreover, the results for model fitness yielded from AMOS were as follows: $\chi 2/df = 2.487$, RMSEA = 0.060, RMR = 0.034, GFI = 0.913, AGFI = 0.890, CFI = 0.932, IFI = 0.933, NFI = 0.903, RFI = 0.907, and NNFI = 0.923. All values met the threshold suggested by Hu and Bentler [83]. It can therefore be concluded that the proposed model reasonably explained the collected data.

**Table 3.** Hypothesis testing results for the three types of sustainable apparel.

| Hypothesis | Path | Rental Apparel | Second-Hand Apparel | Recycled Apparel | Overall Sustainable Apparel |
|---|---|---|---|---|---|
| H1 | EC→PV | Supported (+) | Supported (+) | Supported (+) | Supported (+) |
| H2 | EC→PR | Not supported | Not supported | Not supported | Not supported |
| H3 | PV→SAPI | Supported (+) | Supported (+) | Supported (+) | Supported (+) |
| H4 | PR→SAPI | Supported (−) | Supported (−) | Not supported | Supported (−) |

EC: Environmental consciousness; PV: Perceived value; PR: Perceived risk; SAPI: Sustainable apparel purchase intention; Overall sustainable apparel covers the three types of sustainable apparel. Significance level: $p < 0.05$; $p < 0.01$; $p < 0.001$.

### 4.3. Moderating Effects of Generational Cohorts

The bootstrapping method was performed using SPSS Process Macro to examine if generational cohorts moderated the relationship among environmental consciousness, perceived value, perceived risk, and sustainable apparel purchase intentions. Table 4 provides the hypothesis testing results for H5 and H6. Regarding overall sustainable apparel, a moderating effect of generational cohorts was found in the relationship between environmental consciousness and perceived value ($\beta = -0.140$, CI = $-0.2683{\sim}-0.0120$), indicating that the relationship between environmental consciousness and perceived value was significantly more positive for the younger generations compared to Generation Xers. A moderating effect of generational cohorts was found in the relationship between environmental consciousness and perceived risk ($\beta = -0.313$, CI = $-0.4827{\sim}-0.1445$), indicating that the younger generations would show stronger negative associations between environmental consciousness and perceived risk than would Generation Xers. Thus, H5 was fully supported. Additionally, results also show that generational cohorts moderate the influence of perceived risk on sustainable apparel purchase intention ($\beta = -0.161$, CI = $-0.2889{\sim}-0.0325$). That is, younger generations would show stronger negative associations between perceived risk and sustainable apparel purchase intention than would Generation Xers. However, the moderating effect of generational cohorts on the relationship between perceived value and sustainable apparel purchase intention was not significant ($\beta = 0.018$, CI = $-0.2159{\sim}0.0002$). H6 was therefore partially supported.

**Table 4.** Hypothesis testing results for the moderator.

| Hypothesis | Path | Rental Apparel | Second-Hand Apparel | Recycled Apparel | Overall Sustainable Apparel |
|---|---|---|---|---|---|
| H5 (a) | EC→PV (EC × GC) | Supported (−) | Supported (−) | Supported (−) | Supported (−) |
| H5 (b) | EC→PR (EC × GC) | Supported (−) | Supported (−) | Supported (−) | Supported (−) |
| H6 (a) | PV→SAPI (PV × GC) | Not supported | Not supported | Not supported | Not supported |
| H6 (b) | PR→SAPI (PR × GC) | Supported (−) | Supported (−) | Not supported | Supported (−) |

EC: Environmental consciousness; PV: Perceived value; PR: Perceived risk; SAPI: Sustainable apparel purchase intention; GC: Generational cohorts; Overall sustainable apparel covers the three types of sustainable apparel. Significance level: $p < 0.05$; $p < 0.01$; $p < 0.001$.

## 5. Discussion

This study set out to examine the relationship between environmental consciousness and sustainable apparel purchase intention and to determine whether generational differences moderate the relationship. We divided current sustainable apparel into three types (rental clothing, second-hand clothing, and recycled clothing) and targeted Generations X, Y, and Z. In terms of Hypotheses 1 and 3, we tested the relationship between environmental consciousness, perceived value, and sustainable apparel purchase intention, and found that environmental consciousness strongly affected sustainable apparel purchase intention through perceived value. These results are consistent with studies that examined the influence of environmental concerns on consumers' value perceptions and purchase intentions, specifically those conducted by Hasan et al. [39] and Baek and Oh [45], who found that environmental concerns and perceived values were directly related to consumer decision-making on purchasing eco-friendly clothing. A possible explanation for this might be that environmental consciousness is viewed as an important component of the consumer decision-making process in the context of sustainable consumption [34,35]. That is, the more environmentally conscious the consumers, the stronger their value perceptions and behavioral intentions become regarding sustainable apparel goods.

As for Hypotheses 2 and 4, we tested the relationship between environmental consciousness, perceived risk, and sustainable apparel purchase intention. Surprisingly, this study did not find a significant association between environmental consciousness and perceived risk. In other words, high environmental consciousness does not lead to a lesser risk perception of sustainable clothing. This inconsistency may be due to the growing level of environmental awareness, and the increasing number of fashion brands and retailers committed to providing services and products that combine environmental awareness with quality and performance [5–7]. In terms of the relationship between perceived risk and sustainable apparel purchase intention, the results support a negative direct effect of perceived risks relating to rental and second-hand clothing on purchase intention; however, no evidence of a remarkable effect of perceived risks relating to recycled clothing was detected. A possible explanation for this might be that rental and second-hand clothing may be perceived to be more unsanitary than products made of recycled materials [12,59]. These results reflect those of Yoo et al. [56] who also found that the risks perceived by buyers differed according to the product characteristics when purchasing sustainable fashion products.

For Hypotheses 5 and 6, we tested the moderating effects of generational cohorts in all paths and found these cohorts weakened the effect of environmental consciousness on perceived value, and perceived risk and the effects of perceived risk on sustainable apparel purchase intentions. The results revealed that Generation Zers would show stronger associations between belief, attitude, and intention compared to Generation Yers and Xers, presenting evidence of moderation by generational cohorts. These results are in line with

those of previous studies [15,17,75,76]. For instance, Liang and Xu [77] confirmed that the younger generations perceived higher values and held higher purchase intentions regarding sustainable apparel products than their older counterparts. In summary, although not all the hypotheses were confirmed, the empirical results strongly support a positive effect of the environmental consciousness and perceived value on sustainable apparel purchase intention, and a negative effect of the perceived risk on sustainable apparel purchase intention. These results imply that the consumers' willingness to purchase sustainable clothing is significantly related to environmental consciousness and valence perceptions. Additionally, the findings of this study suggest that consumers' sustainable consumption behaviors could be modified by generational cohorts, proving that young consumers are driving the market for sustainable products and services.

## 6. Conclusions

This study was conducted to disentangle the factors determining the consumers' sustainable apparel purchase intentions including rental, second-hand, and recycled clothing, and to explore the moderating effect of generational cohorts. The most obvious finding to emerge from the study is that environmental considerations and perceptions are important factors in explaining consumer sustainable apparel purchase intention, and a moderating effect of generational cohorts exists in the consumers' sustainable apparel purchase intentions. The present study generates theoretical contributions by synthesizing five research dimensions in the sustainable apparel consumption domain. Specifically, this study advances our understanding of sustainable apparel consumption by exploring consumers' environmental consciousness, perceived value, perceived risk, and purchase intention for different sustainable apparel products. Moreover, this study is the first comprehensive investigation of the moderating role of generations in consumers' pro-environmental attitudes and behaviors regarding sustainable apparel products. Overall, the present study provides a deeper insight into the rapidly expanding field of circular fashion consumption and suggests that sustainable apparel purchase intention varies across market segments and generations.

This study has two main practical contributions. Firstly, it can be a guideline for retailers and marketers in the fashion industry to understand the factors to be considered in promoting sustainable apparel. Particularly, enhancing environmental and functional value may be an effective way of facilitating consumers' sustainable apparel buying intentions. Moreover, it is possible to increase buying intention of sustainable apparel products by reducing consumers' risk perception through promoting specific and genuine environmental friendliness in products from eco-friendly brands. Secondly, it provides a deeper insight into how consumers' sustainable shopping attitudes and behaviors can vary across the different generations. For younger generations, marketers should promote the idea of buying less, and produce fashion products that are made with minimal and environmentally friendly resources. Meanwhile, to make older generations more knowledgeable about sustainable apparel products, marketers and policy enforcers should spread information about sustainable issues through legal guidelines, media reports, and public relations in the manufacturing and marketing sectors of companies. However, there are certain limitations that restrict the generalizability of these findings. Firstly, the sample size and the target sample impeded the generalization of the results. A larger sample size is required to improve the generalizability of research findings. This study was conducted in Taiwan, and further research could be conducted in other countries to extend the understanding in sustainable fashion consumption from a global perspective. Secondly, this study relies on data collected using a questionnaire. Future studies should be undertaken to provide an in-depth view regarding the customers' perceptions and behaviors when purchasing sustainable apparel, by utilizing qualitative techniques such as open-ended survey questionnaires and interviews. Thirdly, this study does not consider the impact of demographic factors such as gender, monthly discretionary income, and marital status. Future work is needed to validate demographic differences in consumers' environmental awareness and sustainable

behaviors. Finally, unifying the four well-known constructs may not fully elaborate the consumers' sustainable behaviors, so future studies should consider incorporating other psychographic and social factors (e.g., lifestyles, values, subjective norms) associated with behavior regarding green consumption.

**Author Contributions:** Conceptualization, P.-H.L.; Formal analysis, P.-H.L.; Investigation, P.-H.L.; Methodology, P.-H.L.; Data curation, P.-H.L.; Writing—original draft, P.-H.L.; Writing—review & editing, P.-H.L.; Supervision, W.-H.C. All authors have read and agreed to the published version of the manuscript.

**Funding:** This research received no external funding.

**Institutional Review Board Statement:** Not applicable.

**Informed Consent Statement:** Informed consent was obtained from all subjects involved in the study.

**Data Availability Statement:** The data presented in this study are available on request from the corresponding author. The data are not publicly available due to privacy reasons.

**Conflicts of Interest:** The authors declare no conflict of interest.

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
