# Peer review of "Factors That Influence Consumers’ Sustainable Apparel Purchase Intention: The Moderating Effect of Generational Cohorts"

_sustainability, doi:10.3390/su14148950_

Round 1

Reviewer 1 Report

I congratulate the authors on an extremely topical topic, which is dealt with by experts in various forums, international world organizations, national governments and international groupings. Based on the current state of pollution of the planet, this problem will be a problem for future generations, and any proposal to reduce the burden will benefit future generations. It is the fashion industry that contributes greatly to the ever-increasing pollution of clothing production in the context of fashion trends.

I would recommend to define more clearly the individual groups of generations that are included in the research, e.g. defining the years of birth from to, which belong to individual groups, the value ladder typical for the given generation, etc.

Conceptual model - I would recommend describing in more detail so that the individual relationships are clearer in it.

Positively evaluating the sample, which offers uniformity, from each sample is about the same number of respondents, which creates space for deduction of proposals.

The list of references is really broad-spectrum, it positively evaluates the use of very current sources, it confirms the authors' view and their ability to correctly use secondary sources, raise research questions and connect them with primary research.

Reviewer 2 Report

Dear Authors,
I’m glad I had the opportunity to revise this interesting work, which aims at identifying Factors that influence consumers’ sustainable apparel purchase intention: The moderating effect of generational cohorts. To do so, factor analysis was performed. I think the investigated topic is timely and it would be interesting for the readers of this Journal and the adopted method is coherent with your aim. However, some revisions are required.

In part of introduction; I have observed some important issues in the Introduction, which will need great work to be improved. In particular, I suggest adopting a commonly used scheme, structuring this section as follows:
•       Broad theme or topic;
•       Academic and practical importance;
•       Literature summary;
•       Gaps, inconsistencies, controversies to be addressed;
•       Focus of the study and research question(s), here or in Section 2;
•       Applied methods;
•       Main results and contributions;
•       Structure of the article (remainder).

At the moment, some of these parts are a little bit vague (i.e., theoretical and practical relevance, research gap) or missing (i.e., paper’s main results and implications, remainder). Moreover, you should try to better position your paper into the current scientific debate. In this sense, take particular care also to avoid supporting recent trends with old references. I also suggest providing a brief explanation/concept of sustainable apparel purchase intention and how do you linked with Factors that influence consumers’. since the Introduction to help international readers contextualizing your analysis and generalizing its results.

Regarding the Materials and Methods, I think that a better justification for investigating the sector should be provided. What distinctive elements make its investigation interesting? What novel insights can be derived from such analysis? Then, some lines dedicated to describing the advantages of adopting a CFA should be added. In addition to that, I'm also wondering why an EFA has not been performed to ensure the reliability and soundness of the model before conducting the CFA. In addition to that, you should add some lines to explain how data have been collected and for AMOS need 200 sample size minimum. What is your sample size? Can you clear me as I am not able to clear about it. For example, which sampling approach was adopted? How can it ensure the best fit with your research's aim? Were the questions open-ended or not? What variables did you use to measure the different constructs? Did you select them according to previous literature on the topic? 
Fifth, I suggest also giving more attention to developing the discussion of your results, trying to better link it to the current scientific debate and the literature leveraged in Section 2. Reading this paragraph, I still cannot capture the distinctive elements that support the relevance of your study. Similarly, the theoretical and practical implications of your work, as well as its limitations and possible further developments, should be added clearly in the Conclusion paragraph or in a separate section of the paper. Unfortunately, there are no novelty of the researh. 

Please add a level of significane i.e., critical value (C.V.) of the each measurement below of every tables. Check other manuscripts how are they mentioning. 

Thank you 

I wish your success

Reviewer 3 Report

The article is written in an appropriate way.

Abstract. The authors clearly state the objective of this research, presenting its results and conclusions.

Introduction. The authors briefly present the importance of the research topic, but the structure of the paper should be described.

Methods. Explain better the platform used to apply the questionnaire. To whom was it addressed? To which country? city? Who had access to the platform? How was the sample chosen?

The conclusions are accurate and reflect the results obtained, but the conclusions should be related with literature review, and it should be presented a comment to the objective. If the objective was achieved or not.

Round 2

Reviewer 2 Report

The authors made substantial corrections. Rather: 

1. In methods part, Please clarify why did you choose the methods with the citations. 

2. What is your population size? Which sampling framework did you used? 

3. Results 4.1 sections; illustreates table variables with value and compared from highest to lowest with the varaibles value. 

4. Our study in conclusion part- delete our/we/I  in the writing. 

5. Identigy the theoretical contribution of the study.

Good luck!
